# Brazilian Adults’ Attitudes and Practices Regarding the Mandatory COVID-19 Vaccination and Their Hesitancy towards Childhood Vaccination

**DOI:** 10.3390/vaccines10111853

**Published:** 2022-11-01

**Authors:** Edson Zangiacomi Martinez, Miriane Lucindo Zucoloto, Vânia Pinheiro Ramos, Carla Daiane Costa Dutra, Giselle Juliana de Jesus, Arinete Véras Fontes Esteves, Isabella Schroeder Abreu, Monica Augusta Mombelli, Roberta Alvarenga Reis, Marilia Marcondes Campoamor, Wanderson Roberto da Silva, Claudia Benedita dos Santos

**Affiliations:** 1Ribeirão Preto Medical School, University of São Paulo (USP), Ribeirão Preto 14049-900, Brazil; 2Nursing Department, Federal University of Pernambuco (UFPE), Recife 50670-901, Brazil; 3Department of Health Science, State University of Santa Cruz (UESC), Ilhéus 45662-900, Brazil; 4Ribeirão Preto College of Nursing, University of São Paulo (USP), Ribeirão Preto 14049-900, Brazil; 5Nursing Department, Federal University of Amazonas (UFAM), Manaus 69077-000, Brazil; 6Nursing Department, Midwestern State University (Unicentro), Guarapuava 85040-080, Brazil; 7Latin American Institute of Life and Nature Sciences, Federal University of Latin American Integration (UNILA), Foz do Iguaçu 85866-000, Brazil; 8Preventive and Social Dentistry Department, Faculty of Dentistry, Federal University of Rio Grande do Sul (UFRGS), Porto Alegre 90035-004, Brazil; 9Faculty of Nursing, University of Rio Verde (Unirv), Goianesia 76380-000, Brazil; 10Graduate Program in Nutrition and Longevity, School of Nutrition, Federal University of Alfenas (UNIFAL-MG), Alfenas 7130-001, Brazil

**Keywords:** COVID-19, vaccine hesitancy, children, paediatrics, public health

## Abstract

Background: This study investigated the attitudes and practices of Brazilian adults regarding the mandatory COVID-19 vaccination and their hesitancy towards the vaccination of children. Methods: Between March and May 2022, Brazilian adults answered an online questionnaire distributed through social media. The SAGE-WG questionnaire was adapted to measure hesitancy to the vaccination of children. Results: Of the 1007 participants, 67.4% believed that adult COVID-19 vaccination should be mandatory. Just over half of the participants (51.5%) believed that parents and/or guardians should decide if their children should be vaccinated against COVID-19 or not and 9.1% were unsure. Individuals who were younger, non-religious and had higher awareness of COVID-19 risks and critics of the federal government’s performance in combating the pandemic were more likely to agree with mandatory adult vaccination. However, less agreement among parents and/or guardians concerning children’s vaccination was observed, with lower scores for hesitancy to the vaccination of children. Conclusion: In Brazil, there is still far from a consensus on mandatory COVID-19 vaccination for adults and a significant proportion of the population believes that parents and/or guardians should be free to decide on their children’s vaccination. These views are associated with age, religion, knowledge of COVID-19 risks and political inclination.

## 1. Introduction

Vaccination is one of the outstanding achievements in public health as it contributes to decline in mortality and morbidity from various infectious diseases [1]. However, some outbreaks of vaccine-preventable diseases, including measles, polio, and pertussis, have occurred in several developed countries which have been mainly associated with groups of individuals with unsatisfactory vaccination coverage or who were unvaccinated [2,3,4,5,6]. Lack of confidence in vaccines is a threat to the success of vaccination programs [1].

According to the SAGE Working Group on Vaccine Hesitancy, vaccine hesitancy refers to delay in accepting or refusing a vaccine despite its availability [7]. For Dubé et al. [1], vaccine hesitancy can be understood conceptually as an individual behavior influenced by many factors, including knowledge and past experiences. Hesitation is also the result of broader influences and should be analysed in the historical, political and sociocultural contexts in which vaccination occurs. It includes trust in the system that provides vaccines, in health professionals who recommend and administer vaccines, in policymakers who decide on vaccination programs and in the different types of information on vaccines conveyed in the media [1]. According to Dubé et al. [8], individuals who demonstrate vaccine hesitancy form very heterogeneous groups and individual attitudes and behaviors towards vaccination do not correspond to a simple dichotomy between acceptance and rejection. Therefore, reducing those who are hesitant to “antivaxxers” is incorrect. While some individuals may refuse all vaccines, others may reject some but accept others. Similarly, some individuals may accept recommended vaccines although they may feel insecure about allowing their children to be vaccinated.

The first reported case of the disease caused by the SARS-CoV-2 coronavirus (COVID-19) in Brazil occurred on 26 February 2020, in São Paulo [9]. On 17 January 2021, the National Health Surveillance Agency (*ANVISA*) authorized the emergency use of two vaccines against the disease in the country, and soon afterwards a nurse working at the Intensive Care Unit of the Emílio Ribas Institute (São Paulo) became the first person vaccinated in the national territory [10]. Despite the scientific evidence supporting childhood vaccination, as highlighted by ANVISA, the federal government decided to hold a public consultation before deciding whether to include children aged 5 to 11 years in the national vaccination campaign [11]. The public consultation was open between 23 December 2021 and 2 January 2022 and was responded to by almost 100,000 individuals. The majority responded that vaccination should not be compulsory.

Nonetheless, in December 2021, ANVISA authorized the vaccination of children aged 5 to 11 old. Thus, on 14 January 2022, 15 children became the first to be immunized in an event promoted by the state government of São Paulo [12]. At least 15 Brazilian Federative Units began vaccination on 15 January 2022. Although Brazil had recorded a cumulative total of more than 27,000 cases of COVID-19 as of February 2022 and more than 630,000 deaths due to the disease, reports of individuals refusing to receive the vaccine or even being against vaccination of their children were common in the media, which was mainly accounted for by the dissemination of fake news on social media [13,14]. In a national web survey using data collected between November 2020 and January 2021, it was found that 30% of the participants showed some type of hesitation about receiving the COVID-19 vaccine [15]. Based on information released by the State Health Secretariats, some media reports published in June 2022 showed that 40% of 5- to 11-year-old children in Brazil had not received the first dose of the COVID-19 vaccine [16]. Although there may have been a delay in the National Immunization Program Information System in completing the data collection process, this figure was below expectations, evidencing slow progress in the vaccination of children in Brazil.

The objective of the present study was to identify variables associated with attitudes and practices regarding mandatory vaccination for COVID-19 in adults and hesitancy towards the vaccination of children. Brazilians aged 18 years and older were invited to participate in the study. We consider that hesitancy is related to an individuals’ decision to allow their children to be vaccinated and the attitudes towards this issue in the community where the participants live. Therefore, participation in the study was independent of whether the respondent had children under 12 years old for whom they were responsible.

## 2. Materials and Methods

### 2.1. Study Design and Settings

This was a cross-sectional study using electronic data collection (open web survey). The *Checklist for Reporting Results of Internet E-Surveys* (CHERRIES) was used to ensure the quality of the data collected and the reliability of the findings [17]. Data collection was based on the Research Electronic Data Capture (REDCap) platform, a secure web-based software platform designed to support data capture for research studies [18]. Invitations for participation in the study, along with a link to the questionnaire, were distributed on social networks (i.e., Facebook, Twitter, Instagram, and WhatsApp) and the mainstream media also helped to publicize the study. The survey was disseminated in all regions of Brazil (i.e., North, Northeast, South, Southeast and Center-West) in an effort to obtain a geographically representative sample of the population of interest. Participation in the survey was voluntary and no compensation was provided. Data were collected from participants between March and May 2022 based on the following inclusion criteria: Participants were required to be aged 18 years or older, to be living in Brazil and to be able to understand the Portuguese language. Participants who left the survey without finishing the questionnaire were not included in the final sample.

### 2.2. Sample Size Determination

The sample size was determined based on a prevalence of 0.5 for hesitancy towards vaccination of children in the population, a confidence coefficient of 0.95 and an absolute error of 0.04. A prevalence of 0.5 represents the highest level of variability in a population and is often used to determine a more conservative sample size; that is, a value typically larger than the true variability in the population characteristic was used in the sample calculation [19]. By assuming an infinite population, a sample size of 600 participants was inferred to be needed to address the research question. It was decided that the on-line questionnaire would be available for completion within two months, though the time could be extended if less than 600 individuals responded to it.

### 2.3. Ethical Issues

The Research Ethics Committee of the Ribeirão Preto Medical School Hospital of the University of São Paulo approved the present study according to protocol number CAAE:56391422.0.0000.5440. The methods used followed the guidelines provided by the Brazilian Research Ethics Commission (Circular Letter 1/2021-CONEP/SECNS/MS) for research procedures including any stage of the research in a virtual environment. The first page of the on-line questionnaire contained an informed consent form which described the objectives and the confidential and voluntary nature of the study. After reading it, the potential respondents could choose whether to take part in the study. Participants were informed that they could withdraw from the study at any time. The first question asked whether the volunteer was 18 years or older. In the case of a negative answer, the survey was terminated. Furthermore, all the questions included the option “I prefer not to answer” in the response categories, implying that the participants were not compelled to answer any question that might make them feel uncomfortable.

### 2.4. Variables

The online questionnaire was developed by the researchers. It included questions relating to sociodemographic variables, such as gender, age, education level and region of residence in Brazil (i.e., North, Northeast, Center-West, Southeast or South). The questions on religion included religious affiliation and how the participant perceived their religiosity (with possible responses ranging from very religious to non-religious). Participants were asked if they had ever had COVID-19, if they had received a vaccine, and whether they thought they were at risk for the disease [20]. Views about the COVID-19 vaccination were evaluated based on the following questions: “Should the vaccine for COVID-19 in adults be mandatory?” and “Should parents and guardians be free to decide on whether their children will receive vaccinations?” [21].

An adaptation of the SAGE-WG questionnaire [22], originally administered to parents and caregivers, was used to measure hesitancy towards the vaccination of children. The SAGE-WG questionnaire was developed based on previously validated instruments [21,23] and did not refer to the vaccine for COVID-19 but to childhood vaccination in general. A Portuguese version of the SAGE-WG questionnaire was presented by Sato [6]. For the purposes of the present study, we adapted the SAGE-WG questionnaire for COVID-19 vaccination according to the ten items presented below:The vaccine for COVID-19 is important for children’s health.The vaccine for COVID-19 can prevent a child from developing the disease.Getting a child vaccinated for COVID-19 is important for the health of other children in the neighborhood or in the same school.The COVID-19 vaccine provided by the *SUS* is beneficial for all children, even those without any disease or health problems.The vaccine for COVID-19 carries more risks than vaccines used for other diseases (such as measles, polio and others).The information given by the *SUS* about the COVID-19 vaccine for children is reliable.Getting children vaccinated can prevent adults living with them from getting COVID-19.It is important to follow the recommendations that the *SUS* providers give about COVID-19 vaccination for children.I am concerned about serious adverse reactions the COVID-19 vaccine may cause to children.Children should get the COVID-19 vaccine, even if the number of cases of the disease is small.

The abbreviation *SUS* in items 4, 6, and 8 refers to the Brazilian Unified Health System (in Portuguese, *Sistema Único de Saúde*), one of the largest and most complex public health systems in the world. A five-point response scale was used (strongly disagree = 4 points; disagree = 3 points; neither agree nor disagree = 2 points; agree = 1 point; strongly agree = 0 points). The sum of the scores of each item, which were reversed for items 5, 9 and 10, yielded the overall score. The overall score ranged from 0 to 40 points— the higher the score, the higher the hesitation to allowing children to be vaccinated.

### 2.5. Statistical Analysis

Qualitative variables were described according to absolute and relative frequencies, with simultaneous 95% confidence intervals (S95%CI) for multinomial proportions obtained using the method proposed by Sison and Glaz [24] implemented in R software with the Multinomial CI package. Associations between the respondent’s views on mandatory vaccination and whether parents and guardians should be free to decide on whether their children should be vaccinated against COVID-19 and variables of interest were analysed using Pearson’s chi-squared test with *p*-values calculated by Monte Carlo simulation with B = 5000 replicates [25]. Cramér’s V coefficients were used as effect size measures, where values below 0.10 indicate negligible association, between 0.10 and below 0.20 indicate weak association, between 0.20 and below 0.40 indicate moderate association, and between 0.40 and 0.60 indicate a relatively strong association [26].

The associations between the scores obtained from the adapted SAGE-WG questionnaire and variables of interest were assessed by linear regression models. The scores were log-transformed to normalize the residuals of the models and to stabilize the within-group variances. In addition, gender and age group were used as covariates to adjust for their potential confounding effects. Given the well-known limitations of *p*-values [27], omega-squared statistics were calculated to indicate the magnitude of associations obtained [28]. According to Cohen, statistical values close to 0.01, 0.06 and 0.14 should be interpreted as small, medium and large effects, respectively [29]. Omega-squared statistics were obtained using the “effectsize” package of the R language.

## 3. Results

Initially, 1072 individuals accessed the online questionnaire. The completion rate was 1011/1072 = 94.3%. Four individuals were excluded for not living in Brazil. Thus, 1007 participants were included in the study. Table 1 compares the distribution of the participants by gender, age group, educational level and Brazilian region to the profile of the Brazilian population according to *IBGE*. Women, individuals living in the southern region and those who had completed higher education were over-represented in our sample, whereas individuals aged 61 or older were under-represented. 

Among the study participants, 677 (67.4%; 95%CI: 64.5% to 70.4%) believed that vaccination for COVID-19 for adults should be mandatory, 284 (28.2%; 95%CI: 25.4% to 31.3%) were opposed to such a policy, and 44 (4.4%; 95%CI: 1.5% to 7.4%) were undecided. Two participants did not answer this question. Table 2 shows that there was no evidence of an association between the belief that the vaccine should be mandatory and variables such as gender, educational level, and region of residence. There was a higher frequency of respondents believing that the vaccine should be mandatory for younger individuals (18–24 years old; 84.7%) and those who considered themselves not very or not at all religious. Evangelicals and protestants were those who most disagreed with mandatory vaccination (38.6% and 39.4%, respectively). Unsurprisingly, all respondents who had not been vaccinated for COVID-19 thought that vaccination should not be mandatory, whereas 74% of those who had received all vaccine doses believed in mandatory vaccination. There were higher percentages of individuals refusing to take the vaccine on a mandatory basis among respondents who did not perceive the risks of COVID-19 to be significant and among those who had already had the disease or were unsure about it. Table 2 also shows that a positive evaluation of the actions taken by the Brazilian federal government regarding the fight against COVID-19 was associated with disagreement with compulsory vaccination.

Just over half of the participants (51.5%; 95%CI: 48.4% to 54.7%) believed that parents and guardians should be free to decide on whether their children should be vaccinated against COVID-19 and 9.1% were unsure about this. Four participants did not answer this question. The percentage of participants in favor of freedom of choice was higher among those who had children aged between 5 and 11 years than among those without children in this age group (62.0% and 47.5%, respectively). The results shown in Table 3 indicate that this view was not associated with gender, education or region of residence, but older individuals tended to be more in favor of freedom of choice regarding vaccination for children. This defense of freedom of choice was more frequent among individuals perceiving themselves as more religious, among those already vaccinated, those not perceiving the risk of COVID-19, those approving the Brazilian government’s actions to combat the disease, and those believing that the disease will finally be controlled.

Figure 1 shows the frequencies of answers to the adapted SAGE-WG questionnaire. Most respondents agreed or agreed strongly that the vaccine for COVID-19 was important for children’s health (76.7%), that the vaccine can prevent a child from developing the disease (66.2%), and that getting a child vaccinated for COVID-19 was important for the health of other children in the neighbourhood or in the same school (76.8%). However, about half of the respondents (51.9%) agreed or strongly agreed that they were concerned about serious adverse reactions the COVID-19 vaccine might cause to children.

Table 4 shows the means and standard deviations for the scores obtained from the adapted SAGE-WG questionnaire according to the variables of interest. The ANOVA and omega-squared effect sizes suggest that hesitancy to the vaccination of children was not associated with gender or region of residence but indicated that it tended to increase with age. Participants who considered themselves to be very religious tended to have a higher hesitancy to the vaccination of children, with the averages being higher among evangelicals and protestants and lower among atheists and individuals with no religion. As expected, individuals who had not received the vaccine for COVID-19 tended to have a higher hesitancy to the vaccination of children. When only those participants with children aged 5 to 11 years were considered, vaccine hesitancy was found to be significantly higher among those with unvaccinated children compared to those with vaccinated children. High mean scores in the adapted SAGE-WG questionnaire were also associated with low perceived risk for the disease, good ratings of the federal government’s performance in combating COVID-19, and belief that the disease will finally be controlled.

## 4. Discussion

The results indicate that among the participants, 67.4% (95%CI: 64.3% to 70.3%) believed that vaccination against COVID-19 for adults should be mandatory and 51.5% (95%CI: 48.4% to 54.7%) believed that parents and guardians should be free to decide on whether their children should be vaccinated against COVID-19. Studies in different parts of the world have investigated individual’s views on vaccination against COVID-19. A study carried out in June and July 2020 in Germany showed that about half of the residents were in favor of a policy of mandatory vaccination, whereas the other half were against [30]. This policy was more likely to be rejected by women and favored by older individuals. There was no evidence of an association between opinion on this policy and the respondents’ political orientation. In a cross-sectional on-line survey involving a representative sample of the French population conducted in May 2021, 43% of the respondents were in favor of mandatory COVID-19 vaccination, 41.9% were opposed to such a policy and 15.1% were undecided [31]. In this survey, the 18–24 and 25–34 year age groups were significantly more opposed to mandatory vaccination than the 75 years or older group. In addition, individuals supporting far-left and green parties were more likely to be opposed to mandatory COVID-19 vaccine [31]. A community-based survey carried out in Portugal from September 2020 to January 2021 showed that refusal to take COVID-19 vaccines was associated with a worse evaluation of government measures to respond to the pandemic [32]. A web survey including 2697 respondents from the US, Canada and Italy showed that individuals who did not believe their governments had responded appropriately to the pandemic were more likely to be vaccine-hesitant [33].

Other researchers have found a link between COVID-19 vaccine hesitancy and negative evaluation of a government’s measures [34,35]. However, in contrast to these findings, our study results indicated that individuals who had a positive evaluation of the federal government’s performance in fighting COVID-19 tended to disagree with mandatory vaccination, believing that parents should be free to decide on whether their children should be vaccinated and expressing higher hesitation towards vaccination. This can be readily explained by the fact that, during the COVID-19 pandemic, the Brazilian government adopted a radical stance of right-wing populism by minimizing the severity of the disease and discrediting the vaccines [36,37]. The Brazilian president insisted on promoting the use of hydroxychloroquine and ivermectin for prevention and treatment of COVID-19 instead of encouraging vaccination of the population, even though these drugs have been shown to be ineffective in treating the disease [38,39,40]. Out of party loyalty or a populist approach, many other politicians advocated the use of dubious pharmacological interventions for COVID-19. In this regard, a German study investigating what affected how citizens responded to mandatory vaccination during the COVID-19 pandemic showed that respondents tended to adjust their position according to the views of their most preferred political party [41]. This partly explains our results.

Aside from political beliefs, our results also showed a relationship between religiosity and vaccination reluctance. Based on the adapted SAGE-WG questionnaire, we observed that non-religious individuals, atheists and individuals who had no religion but believed in God had lower vaccine hesitancy scores, tended to be in favour of mandatory adult vaccination and did not think parents should be free to decide on whether their children should be vaccinated. Various authors have shown that religious issues play an important role in a person’s attitudes, beliefs, and decisions [42,43]. However, the way in which religious affiliation and beliefs shape attitudes and behaviours towards vaccination can vary greatly from one country to another, according to social and cultural characteristics. During the COVID-19 pandemic, it was noted that there was a close connection between conservative religious leaders and the Brazilian president, which also contributed to discrediting the vaccine and promoting ineffective treatments [44,45]. From the beginning of the period of social isolation, well-known Brazilian pastors of neo-pentecostal churches refused to suspend public worship services and disseminated messages through their blogs and social networks calling for their followers not to fear the virus, as God would protect those who had faith [46]. Influential personalities associated with spiritism, a religion representing the third largest religious segment in Brazil, also declared their support for the federal government and contributed to the dissemination of fake news, scientific denialism and misinformation about COVID-19 [47]. However, while our findings show that evangelicals and protestants had higher mean vaccine hesitancy scores, according to the adapted SAGE-WG questionnaire (17.58 and 18.67, respectively), spiritists had a mean score of 12.95, similar to that of individuals who had no religion but believed in God (12.91). This highlights that the mechanisms underlying the relationship between religiosity and vaccination reluctance are complex and can be moderated by a variety of factors, including religious involvement, sectarianism, adherence to social norms of the religious group and the political interests of religious leaders and influencers. Further studies are needed to assess the nature of these relationships in the Brazilian population.

This study has some limitations that need to be considered. First, this is a cross-sectional study in which causal interpretations cannot be made. Second, some potentially important variables, such as family type, family structure, marital status, and number of children, were not included in the survey. Third, because our results did not distinguish between hesitation and rejection, we were unable to determine how many participants were genuinely hesitant and how many were simply refusers. Fourth, due to the urgency for research concerning hesitancy about childhood vaccination in Brazil, the adapted SAGE-WG questionnaire used in this study was not validated for the Brazilian population. We adapted the Portuguese language version of the SAGE-WG questionnaire proposed by Sato [6] by specifying the words “vaccine” and “vaccinated” in the context of COVID-19, but a content validation of its items was not performed. Fifth, we generalized the respondents’ evaluation of the federal government’s performance in combating the COVID-19 although Brazil is a federation, in which the federative units (states) have some autonomy to decide on vaccination. Therefore, a participant may have a positive evaluation of the federal government’s performance but a negative one of the state government’s performance (or vice versa), but this was not identified in the present study. Sixth, this study relied on self-reported answers, and the results might, therefore, include social desirability bias. Moreover, those who participate in online surveys may misrepresent their responses. Seventh, personal data were not collected in order to maintain respondents anonymity, which could have led to a bias, such as participants answering the questionnaire more than once. However, we consider this issue unimportant compared to the importance of encouraging spontaneous answers from respondents. Eighth, the sampling method might have increased the possibility of participant self-selection. In addition, on-line surveys have low response rates [48]. Table 1 shows that our sample was mostly composed of women with high education levels, which is the case in other Brazilian web surveys based on convenience samples targeting a broad population [49,50,51]. The most vulnerable individuals with low schooling levels and living in poverty may express less willingness to be vaccinated [15]. Despite these problems, Weigold et al. [52] showed that self-report survey-based instruments can generally be administered through the Internet with good results. Although our results are valid only for particular groups, they suggest that understanding people’s hesitancy to the vaccination of children in Brazil requires the assessment of the political-religious characteristics of the population, including variables such as risk perception, gender, age and age group. More complex sampling schemes, such as those based on chain referral sampling techniques [53], should be used in future studies.

## 5. Conclusions

Our findings show that, for the population represented by the study participants, there was far from a consensus on the mandatory COVID-19 vaccination of adults, as nearly half of participants believed that parents and guardians should be free to decide whether their children should be vaccinated, whereas the other half believed the contrary. The current period of political turbulence in Brazil, combined with religious leaders’ influence on the control of COVID-19, has a major impact on individual’s decisions on whether to be vaccinated and whether to have their children vaccinated as well. Vaccine acceptance among the general public has an essential role in successfully controlling the pandemic and studies on population characteristics associated with vaccine hesitancy are essential for planning strategies to prevent the perpetuation of the COVID-19 pandemic in Brazil.

## Figures and Tables

**Figure 1 vaccines-10-01853-f001:**
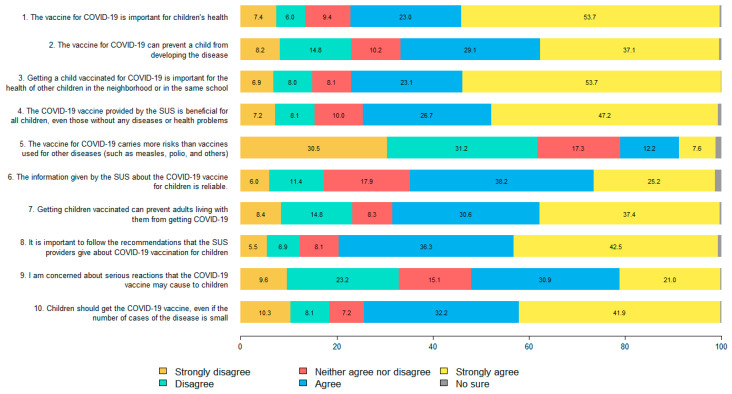
Frequencies of answers to the adapted SAGE-WG questionnaire. The abbreviation SUS in the questions 4, 6, and 8 refers to the Brazilian Unified Health System (in Portuguese, *Sistema Único de Saúde*).

**Table 1 vaccines-10-01853-t001:** Characteristics of the study participants and comparisons with the profile of the Brazilian population.

Variable	Participants *n* (%) ^1^	S95%CI for Proportions	Brazilian Population (%) ^2^
Gender (*n* =1001)			
Female	725 (72.4)	(69.6, 75.2)	51.8
Male	276 (27.6)	(24.8, 30.4)	48.2
Age group (years) (*n* =1004)			
18–24	177 (17.6)	(14.7, 20.7)	14.9
25–30	136 (13.6)	(10.7, 16.6)	12.7
31–35	138 (13.8)	(10.9, 16.8)	10.6
36–40	145 (14.4)	(11.6, 17.5)	10.9
41–50	207 (20.6)	(17.7, 23.7)	18.3
51–60	132 (13.1)	(10.3, 16.2)	14.8
61 or older	69 (6.9)	(4.0, 9.9)	17.8
Education level (*n* =1006)			
No schooling or incomplete elementary school	2 (0.2)	(0, 3.1)	38.7
Completed elementary school or incomplete high school	9 (0.9)	(0, 3.8)	12.5
Completed high school or incomplete higher education	293 (29.1)	(26.2, 32.0)	31.4
Completed higher education	702 (69.8)	(66.9, 72.7)	17.5
Brazilian Region (*n* =1006)			
Southeast	335 (33.3)	(30.1, 36.5)	43.3
Northeast	234 (23.2)	(20.1, 26.5)	26.3
South	201 (20.0)	(16.8, 23.2)	14.8
North	128 (12.7)	(9.5, 16.0)	7.8
Central-West	109 (10.8)	(7.6, 14.1)	7.8

S95%CI: simultaneous 95% confidence intervals for proportions (Sison and Glaz method) ^1^. Totals differ between variables because of missing observations ^2^. According to Brazilian Institute of Geography and Statistics (IBGE).

**Table 2 vaccines-10-01853-t002:** Distribution of participants regarding views about mandatory vaccination for COVID-19 in adults.

		Should the Vaccine for COVID-19 in Adults Be Mandatory?			
	Total ^1^	Yes *n* (%)	No *n* (%)	Not Sure *n* (%)	Effect Size ^2^ (*p* Value) ^3^
Gender (*n* = 999)					
Female	723	503 (69.6)	190 (26.3)	30 (4.1)	0.069
Male	276	172 (62.3)	90 (32.6)	14 (5.1)	(0.089)
Age groups (years) (*n* =1002)					
18–24	177	150 (84.7)	22 (12.4)	5 (2.8)	0.125
25–30	136	90 (66.2)	41 (30.1)	5 (3.7)	(<0.001)
31–40	283	182 (64.3)	88 (31.1)	13 (4.6)	
41–60	337	209 (62.0)	110 (32.6)	18 (5.3)	
61 or older	69	44 (63.8)	22 (31.9)	3 (4.3)	
Education level (*n* = 1004)					
No schooling or incomplete elementary school	2	2 (100.0)	0 (0.0)	0 (0.0)	0.057
Completed elementary school or incomplete high school	9	6 (66.7)	3 (33.3)	0 (0.0)	(0.308)
Completed high school or incomplete higher education	292	212 (72.6)	69 (23.6)	11 (3.8)	
Completed higher education	701	457 (65.2)	211 (30.1)	33 (4.7)	
Brazilian Region (*n* = 1005)					
Southeast	335	209 (62.4)	105 (31.3)	21 (6.3)	0.079
Northeast	234	170 (72.6)	58 (24.8)	6 (2.6)	(0.129)
South	201	130 (64.7)	64 (31.8)	7 (3.5)	
North	127	92 (72.4)	29 (22.8)	6 (4.7)	
Central-West	108	76 (70.4)	28 (25.9)	4 (3.7)	
Are you a religious person? (*n* = 969)					
Very religious	148	87 (58.8)	57 (38.5)	4 (2.7)	0.086
Moderately religious	441	295 (66.9)	126 (28.6)	20 (4.5)	(0.025)
A little religious	250	181 (72.4)	56 (22.4)	13 (5.2)	
Non-religious	130	95 (73.1)	30 (23.1)	5 (3.8)	
Have a religion (*n* = 976)					
Catholic	385	265 (68.8)	102 (26.5)	18 (4.7)	0.108
No religion, but believe in God	197	140 (71.1)	47 (23.9)	10 (5.1)	(0.198)
Evangelic	114	65 (57.0)	44 (38.6)	5 (4.4)	
Spiritist	97	70 (72.2)	24 (24.7)	3 (3.1)	
Atheist	63	50 (79.4)	12 (19.0)	1 (1.6)	
Protestant	33	17 (51.5)	13 (39.4)	3 (9.1)	
Spiritualist	31	19 (61.3)	11 (35.5)	1 (3.2)	
Umbandist	17	12 (70.6)	5 (29.4)	0 (0.0)	
Buddhist	6	5 (83.3)	1 (16.7)	0 (0.0)	
Other religions	33	22 (66.7)	11 (33.3)	0 (0.0)	
Have you ever had COVID-19? (*n* = 1004)					
No	415	297 (71.6)	95 (22.9)	23 (5.5)	0.094
Yes	519	344 (66.3)	159 (30.6)	16 (3.1)	(0.001)
Not sure	70	36 (51.4)	29 (41.4)	5 (7.1)	
Have you received a vaccination for COVID-19? (*n* = 995)					
No	37	0 (0.0)	37 (100.0)	0 (0.0)	0.327
Yes, but only one dose	55	6 (10.9)	48 (87.3)	1 (1.8)	(<0.001)
Yes, all doses available	903	670 (74.2)	191 (21.2)	42 (4.7)	
Self-perception of risk (*n* = 1002)					
Very high	75	52 (69.3)	21 (28.0)	2 (2.7)	0.147
High	243	162 (66.7)	68 (28.0)	13 (5.3)	(<0.001)
Low	436	320 (73.4)	103 (23.6)	13 (3.0)	
Very low	161	102 (63.4)	51 (31.7)	8 (5.0)	
No risk at all	19	3 (15.8)	15 (78.9)	1 (5.3)	
Not sure	68	37 (54.4)	24 (35.3)	7 (10.3)	
Do you consider your knowledge of COVID-19 satisfactory? (*n* = 1003)					
Yes	830	546 (65.8)	250 (30.1)	34 (4.1)	0.090
No	92	62 (67.4)	22 (23.9)	8 (8.7)	(0.005)
Not sure	81	68 (84.0)	12 (14.8)	1 (1.2)	
How do you evaluate the federal government’s performance in combating COVID-19? (*n* = 993)					
Very good	116	38 (32.8)	75 (64.7)	3 (2.6)	0.301
Good	163	81 (49.7)	71 (43.6)	11 (6.7)	(<0.001)
Average	193	123 (63.7)	64 (33.2)	6 (3.1)	
Bad	155	117 (75.5)	31 (20.0)	7 (4.5)	
Very bad	366	318 (86.9)	32 (8.7)	16 (4.4)	
Do you agree that COVID-19 will finally be successfully controlled? (*n* = 1001)					
Yes	514	320 (62.3)	176 (34.2)	18 (3.5)	0.133
No	227	158 (69.6)	63 (27.8)	6 (2.6)	(<0.001)
Not sure	260	199 (76.5)	41 (15.8)	20 (7.7)	
Have all your children between 5 and 11 years old received the COVID-19 vaccine? (*n* = 257) ^4^					
Yes	168	120 (71.5)	37 (22.0)	11 (6.5)	0.489
No	89	21 (23.6)	64 (71.9)	4 (4.5)	(<0.001)

^1^. Two participants were not included as they did not answer the question about the vaccine being compulsory. Totals differ between variables because of missing observations ^2^. Cramér’s V coefficient ^3^. *p*-Values computed for a Monte Carlo test with B = 5000 replicates ^4^. Considering 257 participants who declared they had children between 5 and 11 years old.

**Table 3 vaccines-10-01853-t003:** Distribution of participants regarding their views on whether parents and guardians should be free to decide whether to have their children vaccinated against COVID-19.

		Should Parents and Guardians Be Free to Decide Whether Their Children Will Receive Vaccinations?			
	Total ^1^	Yes *n* (%)	No *n* (%)	Not Sure *n* (%)	Effect Size ^2^ (*p* Value) ^3^
Gender (*n* = 997)					
Female	722	366 (50.7)	288 (39.9)	68 (9.4)	0.029
Male	275	148 (53.8)	104 (37.8)	23 (8.4)	(0.662)
Age groups (years) (*n* = 1000)					
18–24	176	63 (35.8)	84 (47.7)	29 (16.5)	0.135
25–30	136	66 (48.5)	58 (42.6)	12 (8.8)	(<0.001)
31–40	281	144 (51.2)	112 (39.9)	25 (8.9)	
41–60	339	205 (60.5)	113 (33.3)	21 (6.2)	
61 or older	68	38 (55.9)	27 (39.7)	3 (4.4)	
Education level (*n* = 1002)					
No schooling or incomplete elementary school	2	2 (100.0)	0 (0.0)	0 (0.0)	0.055
Completed elementary school or incomplete high school	8	5 (62.5)	3 (37.5)	0 (0.0)	(0.397)
Completed high school or incomplete higher education	292	145 (49.7)	113 (38.7)	34 (11.6)	
Completed higher education	700	365 (52.1)	279 (39.9)	56 (8.0)	
Brazilian Region (*n* = 1003)					
Southeast	334	167 (50.0)	139 (41.6)	28 (8.4)	0.078
Northeast	233	105 (45.1)	102 (43.8)	26 (11.2)	(0.146)
South	201	108 (53.7)	77 (38.3)	16 (8.0)	
North	128	79 (61.7)	40 (31.2)	9 (7.0)	
Central-West	107	58 (54.2)	37 (34.6)	12 (11.2)	
Are you a religious person? (*n* = 967)					
Very religious	147	96 (65.3)	40 (27.2)	11 (7.5)	0.183
Moderately religious	440	258 (58.6)	145 (33.0)	37 (8.4)	(<0.001)
A little religious	250	102 (40.8)	117 (46.8)	31 (12.4)	
Non-religious	130	37 (28.5)	82 (63.1)	11 (8.5)	
Have a religion (*n* = 974)					
Catholic	386	212 (54.9)	141 (36.5)	33 (8.5)	0.194
No religion, but believe in God	196	84 (42.9)	88 (44.9)	24 (12.2)	(<0.001)
Evangelic	115	83 (72.2)	24 (20.9)	8 (7.0)	
Spiritist	95	45 (47.4)	42 (44.2)	8 (8.4)	
Atheist	63	10 (15.9)	45 (71.4)	8 (12.7)	
Protestant	32	23 (71.9)	6 (18.8)	3 (9.4)	
Spiritualist	31	16 (51.6)	12 (38.7)	3 (9.7)	
Umbandist	17	7 (41.2)	9 (52.9)	1 (5.9)	
Buddhist	6	4 (66.7)	2 (33.3)	0 (0.0)	
Other religions	33	15 (45.5)	17 (51.5)	1 (3.0)	
Have you ever had COVID-19? (*n* = 1002)					
No	413	195 (47.2)	184 (44.6)	34 (8.2)	0.067
Yes	519	282 (54.3)	189 (36.4)	48 (9.2)	(0.064)
Not sure	70	39 (55.7)	22 (31.4)	9 (12.9)	
Have you received a vaccination for COVID-19? (*n* = 993)					
No	37	36 (97.3)	1 (2.7)	0 (0.0)	0.197
Yes, but only one dose	55	51 (92.7)	3 (5.5)	1 (1.8)	(<0.001)
Yes, all doses available	901	420 (46.6)	391 (43.4)	90 (10.0)	
Self-perception of risk (*n* = 1000)					
Very high	75	36 (48.0)	37 (49.3)	2 (2.7)	0.109
High	243	123 (50.6)	98 (40.3)	22 (9.1)	(0.009)
Low	435	207 (47.6)	182 (41.8)	46 (10.6)	
Very low	161	88 (54.7)	57 (35.4)	16 (9.9)	
No risk at all	19	15 (78.9)	4 (21.1)	0 (0.0)	
Not sure	67	46 (68.7)	17 (25.4)	4 (6.0)	
Do you consider your knowledge of COVID-19 satisfactory? (*n* = 1001)					
Yes	828	432 (52.2)	327 (39.5)	69 (8.3)	0.066
No	92	51 (55.4)	29 (31.5)	12 (13.0)	(0.066)
Not sure	81	32 (39.5)	39 (48.1)	10 (12.3)	
How do you evaluate the federal government’s performance in combating COVID-19? (*n* = 992)					
Very good	116	101 (87.1)	8 (6.9)	7 (6.0)	0.366
Good	164	124 (75.6)	27 (16.5)	13 (7.9)	(<0.001)
Average	191	121 (63.4)	53 (27.7)	17 (8.9)	
Bad	155	70 (45.2)	55 (35.5)	30 (19.4)	
Very bad	366	91 (24.9)	252 (68.9)	23 (6.3)	
Do you agree that COVID-19 will finally be successfully controlled? (*n* = 999)					
Yes	513	296 (57.7)	171 (33.3)	46 (9.0)	0.117
No	228	114 (50.0)	101 (44.3)	13 (5.7)	(<0.001)
Not sure	258	103 (39.9)	123 (47.7)	32 (12.4)	
Have children between the ages of 5 and 11 (*n* = 990)					
Yes	255	158 (62.0)	78 (30.6)	19 (7.5)	0.127
No	735	349 (47.5)	315 (42.9)	71 (9.7)	(<0.001)
Have all your children between 5 and 11 years old received the COVID-19 vaccine? (*n* = 255) ^4^					
Yes	167	78 (46.7)	72 (43.1)	17 (10.2)	0.433
No	88	80 (90.9)	6 (6.8)	2 (2.3)	(<0.001)

^1^. Four participants were not included because they did not give their opinion on whether parents and guardians should be free to have their children vaccinated against COVID-19. Totals differ between variables because of missing observations ^2^. Cramér’s V coefficient ^3^. *p*-Values computed for a Monte Carlo test with B = 5000 replicates ^4^. Considering 255 participants who declared they had children between 5 and 11 years old.

**Table 4 vaccines-10-01853-t004:** Descriptive statistics for the scores obtained from the adapted SAGE-WG questionnaire, and comparisons between groups. Higher scores suggest higher vaccination hesitance intensity.

	Total ^1^	Mean (SD)	*p* Value ^2^	ω^2^ Statistics ^3^
Gender (*n* = 962)				
Female	698	13.70 (7.88)	0.202	<0.01
Male	264	14.83 (8.84)		
Age groups (years) (*n* = 964)				
18–24	168	11.31 (5.90)	<0.001	0.02
25–30	131	13.85 (7.81)		
31–40	273	13.88 (8.17)		
41–60	326	15.31 (8.74)		
61 or older	66	15.36 (9.33)		
Education level (*n* = 999)				
No schooling or incomplete elementary school	2	14.50 (7.78)	0.010	<0.01
Completed elementary school or incomplete high school	8	12.12 (10.08)		
Completed high school or incomplete higher education	276	13.53 (7.01)		
Completed higher education	680	14.22 (8.58)		
Brazilian Region (*n* = 967)				
Southeast	328	14.63 (8.67)	0.112	<0.01
South	191	14.95 (8.93)		
Northeast	222	12.72 (7.35)		
North	121	13.84 (7.64)		
Central-West	105	13.45 (7.18)		
Are you a religious person? (*n* = 933)				
Very religious	143	16.71 (10.04)	<0.001	0.03
Moderately religious	423	14.65 (7.69)		
A little religious	242	12.19 (6.97)		
Non-religious	125	11.58 (7.81)		
Have a religion (*n* = 939)				
Catholic	374	13.68 (7.79)	<0.001	0.05
No religion, but believe in God	187	12.91 (8.11)		
Evangelic	110	17.58 (8.05)		
Spiritist	92	12.95 (7.50)		
Atheist	61	10.23 (5.82)		
Protestant	30	18.67 (8.40)		
Spiritualist	29	14.62 (9.47)		
Umbandist	17	15.41 (10.15)		
Buddhist	6	13.50 (11.33)		
Other religions	33	14.24 (7.86)		
Have you ever had COVID-19? (*n* = 966)				
No	402	12.79 (7.48)	<0.01	0.02
Yes	497	14.86 (8.46)		
Not sure	67	14.96 (8.96)		
Have you received a vaccination for COVID-19? (*n* = 958)				
No	34	32.79 (4.07)	<0.01	0.21
Yes, but only one dose	51	26.63 (7.85)		
Yes, all doses available	873	12.38 (6.42)		
Self-perception of risk (*n* = 965)				
Very high	72	11.90 (6.25)	<0.01	0.05
High	232	14.12 (7.68)		
Low	427	13.09 (7.51)		
Very low	157	14.37 (9.27)		
No risk at all	16	27.81 (9.94)		
Not sure	61	18.07 (8.54)		
Do you consider your knowledge of COVID-19 satisfactory? (*n* = 965)				
Yes	805	13.98 (8.43)	0.14	<0.01
No	84	15.07 (6.92)		
Not sure	76	13.07 (6.34)		
How do you evaluate the federal government’s performance in combating COVID-19? (*n* = 959)				
Very good	113	23.81 (8.87)	<0.01	0.04
Good	158	18.12 (7.38)		
Average	185	15.29 (7.39)		
Bad	147	12.12 (6.08)		
Very bad	356	8.99 (4.37)		
Do you agree that COVID-19 will finally be successfully controlled? (*n* = 963)				
Yes	493	14.86 (8.64)	0.01	<0.01
No	224	13.69 (8.20)		
Not sure	246	12.40 (6.59)		
Have children between the ages of 5 and 11 (*n* = 957)				
Yes	249	16.42 (9.30)	<0.01	0.02
No	708	13.05 (7.48)		
Have all your children between 5 and 11 years old received the COVID-19 vaccine? (*n* = 249) ^4^				
Yes	167	11.75 (5.92)	<0.01	0.44
No	82	25.93 (7.51)		

SD: Standard deviation ^1^. Totals differ between variables because of missing observations ^2^. *p*-Values from linear regression models with sex and age as covariates, testing the null hypothesis of non-association between each variable and the scores obtained from the adapted SAGE-WG questionnaire ^3^. Omega-squared statistics, where values close to 0.01, 0.06, and 0.14 should be interpreted as small, medium, and large effects, respectively ^4^. Considering 249 participants who declared they had children between 5 and 11 years old.

## Data Availability

The data that support the findings of this study are available from the corresponding author (EZM) upon reasonable request.

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
