# Peer review of "Brazilian Adults’ Attitudes and Practices Regarding the Mandatory COVID-19 Vaccination and Their Hesitancy towards Childhood Vaccination"

_vaccines, 2022, doi:10.3390/vaccines10111853_

Round 1
Reviewer 1 Report
This study tries to identify factors associated with mandatory vaccination for COVID-19 in adults and the hesitancy to vaccinate children. My comments are presented below:
1. The representativeness of the sample is questionable. There are more than 210 million people in Brazil, and yet, the study was only able to recruit about 1,100 participants. The authors did not try to evaluate the representativeness of the sample. I am very concerned about the results and any conclusions derived from this study.
2. The total and sub-total numbers in and across the tables are not consistent. The authors just put a note at the end of Table 1 saying: “Numbers may not sum to total due to missing data” This makes it harder for readers to evaluate what the numbers mean. The authors should explain the evolution of their sample size. They should tell readers how and why there were missing cases in detail so that readers could have a judgment on the reasonableness of how the data were collected and handled.
3. The manuscript was written poorly. Many incoherent sentences can be found throughout the text. For example:
Line 99: Therefore, participation in the study was independent of whether or not the person has 99 individuals under 12 years of age under their responsibility.
Line 169: The total score is given by the sum of points of the items, where the points for items 5, 9, and 10 must be previously reversed
Line 199: people living in the South Region, and those with a complete higher education are overrepresented in our sample
Author Response
Reviewer #1
Thank you for your comments on our manuscript entitled “Brazilian adults' attitudes and practices toward the mandatory COVID-19 vaccination and their hesitancy about childhood vaccination". All the comments were valuable and helpful to the revision and improvement of the manuscript. We have carefully studied the comments and made corrections, which we hope will merit your approval.
This study tries to identify factors associated with mandatory vaccination for COVID-19 in adults and the hesitancy to vaccinate children. My comments are presented below:
- The representativeness of the sample is questionable. There are more than 210 million people in Brazil, and yet, the study was only able to recruit about 1,100 participants. The authors did not try to evaluate the representativeness of the sample. I am very concerned about the results and any conclusions derived from this study.
Answer: Thank you for the opportunity to clarify this important point. Usual strategies of sample size calculation are based on the Central Limit Theorem, which assumes that when a population is repeatedly sampled, the average value of the parameter estimate obtained by those samples is equal to the true population value. The variability of these parameter estimates, obtained from different samples drawn from the population, is the standard error. The larger the sample size, the smaller the standard error and the greater the precision of our estimate. When dealing with finite populations, the sample size calculation based on the standard error uses the called Finite Population Correction Factor (FPC), given by FPC = square-root((N-n)/(N-1)), where N denotes the population size, and n denotes the sample size. The PCF measures the effect of population size on the sample size calculation. If the calculated value for the FPC is close to 1, we can then use conventional sample size calculation formulas for infinite populations, without worrying about population size. Assuming a population of 210 million people, we have FPC = 0.9999976 for a sample of size n = 1000, and FPC = 0.9997619 for a sample of size n = 100000. Therefore, if we consider the context of calculating a sample size for a proportion, we can use the traditional expression n = (1.96^2*p*(1-p))/(d^2), where p is a value for the proportion and d is the absolute error. Based on this expression, we have that the absolute errors for samples of size n = 1000, 2000, 5000, 10000, and 100000 are given by 0.031, 0.022, 0.014, 0.010, and 0.003, respectively. This shows us that, even for a population of 210 million people, a sample size of 1000 is sufficient to provide a small error, and very larger samples do not bring important decreases in absolute error. Note that in periods of elections for president of Brazil, poll institutes traditionally use samples of 2000 to 5000 voters, which, according to these ideas presented, are able to represent the entire population of voters in the country. Using statistical theory, we can argue that our sample size is representative of the Brazilian population. We have added a new subsection in the revised version of the manuscript, with a justification for the sample size adopted.
- The total and sub-total numbers in and across the tables are not consistent. The authors just put a note at the end of Table 1 saying: “Numbers may not sum to total due to missing data” This makes it harder for readers to evaluate what the numbers mean. The authors should explain the evolution of their sample size. They should tell readers how and why there were missing cases in detail so that readers could have a judgment on the reasonableness of how the data were collected and handled.
Answer: Thanks for pointing this out. We clarified this issue in the manuscript.
- The manuscript was written poorly. Many incoherent sentences can be found throughout the text. For example:
Line 99: Therefore, participation in the study was independent of whether or not the person has 99 individuals under 12 years of age under their responsibility.
Line 169: The total score is given by the sum of points of the items, where the points for items 5, 9, and 10 must be previously reversed
Line 199: people living in the South Region, and those with a complete higher education are overrepresented in our sample
Answer: Dear Reviewer, thank you for noting this. We rechecked the manuscript for typos and language errors.
Reviewer 2 Report
www.mdpi.com/journal/vaccines
Brazilian adults' attitudes and practices toward the mandatory vaccination for COVID-19 and the hesitancy about childhood vaccination.
In this study, Martines et al. intended to evaluate the attitudes and practices of Brazilian adults regarding the mandatory vaccination for COVID-19 and the hesitancy to children´s vaccination. This is an interesting study, however, the manuscript would benefit from the following changes/amendments:
1. Abstract: Basic demographic features of the participants should be given.
2. Online surveys have low response rate. The authors should mention this in the manuscript.
3. Please give the exclusion/inclusion criteria for sampling.
4. It is not clear how many individuals were approached and how many respondents did no provide complete information.
5. Please mention the duration of the study.
6.
7. Such type of online survey studies have generally sampling biasness and the attitude of respondents is not serious. Further, there is high error rates. Hence, the data gathered have strong limitations. This should be mentioned in the paper.
8. It is intriguing to see that authors have not gathered/presented data on family type, family structure, marital status and number of children, as these variables are important to consider for the outcome.
Author Response
Reviewer #2
In this study, Martines et al. intended to evaluate the attitudes and practices of Brazilian adults regarding the mandatory vaccination for COVID-19 and the hesitancy to children´s vaccination. This is an interesting study, however, the manuscript would benefit from the following changes/amendments:
We thank the reviewer for kind words. We have revised our manuscript based on the comments as described below.
- Abstract: Basic demographic features of the participants should be given.
Answer: We agree with the reviewer's comment, but we found it difficult to add information in the abstract, given that it cannot be more than 200 words long, according to the journal's rules. We believe that the information currently in the abstract is essential, and it will not be easy to decide what should be removed to add the characteristics of the participants.
- Online surveys have low response rate. The authors should mention this in the manuscript.
Answer: We have, as suggested, included a sentence in the limitations-section of the paper.
- Please give the exclusion/inclusion criteria for sampling.
Answer: Inclusion criteria for the study were being 18 years or older, being a resident of Brazil, and having the ability to understand the Portuguese language. Participants who ended the survey without completing the questionnaire were excluded. We revised the manuscript according to your suggestion and added the inclusion and exclusion criteria in the methods section.
- It is not clear how many individuals were approached and how many respondents did no provide complete information.
Answer: Overall, 1072 people accessed the online questionnaire's starting page; 1011 respondents then started and completed the survey. We eliminated four respondents who did not live in Brazil. We have revised the sentences to facilitate better reading and understanding of this information.
- Please mention the duration of the study.
Answer: We apologize for this mistake. This was corrected. The data was collected between March and May 2022. We have now added a sentence in the “Study design and settings” subsection to include the duration of the study.
- Such type of online survey studies have generally sampling biasness and the attitude of respondents is not serious. Further, there is high error rates. Hence, the data gathered have strong limitations. This should be mentioned in the paper.
Answer: Thank you for pointing this out. This study relied on self-reported answers, and the results might thus include social desirability bias. Moreover, those who participate in online surveys may misrepresent their responses. We have added these commentaries at the end of the paper, among the limitations of the study.
- It is intriguing to see that authors have not gathered/presented data on family type, family structure, marital status and number of children, as these variables are important to consider for the outcome.
Answer: We agree with the reviewer. We designed a relatively short questionnaire to minimize the dropout rate. Consequently, some important variables were not considered. We have added a comment about this problem at the end of the paper, among the limitations of the study.
Reviewer 3 Report
Congratulations - this is a well conducted study and also well written manuscript which however needs some considerations and modifications.
- the authors need to highlight more that the methodology of a social media based survey introduces a selection bias (older people, low income people, probably North and NorthEast Brasil) with lower social media adherence. Likewise, it needs to be made clearer that this is NOT a representative survey.
- What was the attrition between people who were first interested but did not meet the inclusion/ exclusion?
- You describe the Q with respect to children - what was the Q for adults only?
- It is unclear and needs to be justified why the lower age for inclusion in the survey was 18 years and why there is a gap in assessement between 12-18 years (you only talk about kids below 12 yoa and then you have a lower age of 18 y). Adolescents are an important multiplier group and apt to social media. This is a missed opportunity.
- you use the term "children hesitancy". In fact, as you did not ask children, you need to modify this term to something like "parent's hesitancy towards children vaccination"
- It would have been beneficial if the authors had differentiated between hesitancy (which is well explained) and rejection: how many of the "No's)" are indeed hesitaters and how many are plain rejectors
- Tbl 2: it is problematic to single out the Federal government: Brasil as a federal union allows also the states quite some autonomy to set up policies incl vaccination priorities and choice of vaccine (Sao Paulo!).
- There was a survey done by the government about paeds vaccination, The outcome should be mentioned and referenced.
- Whilst the mortality data in Brasil are correctly described and discussed, it is incomplete as the excess mortality data which is similar to ie US or Italy should also be described and referenced.
- line 278: sentence incomplete
- line 316: there are no data in your survey which justify this statement. Pls avoid politicising your work. This is all not black and white - whilst part of the government indeed voiced rethorics against vaccination, at the same time Brasil acquired over 680 Mio doses. And the coverage is among the highest in large size countries. What also did not help was the attitude of some governors and one manufacturer who made public flawed data and criticised organisations such as Anvisa which is perceived internationally as one of the agencies who best mastered the pandemic.
-
Author Response
Reviewer #3
Congratulations - this is a well conducted study and also well written manuscript which however needs some considerations and modifications.
Answer: Thank you for your appreciation, observation and suggestion. We have revised the whole manuscript and made changes where it was necessary. We hope that after this revision, the paper will be deemed fit for publication.
- the authors need to highlight more that the methodology of a social media based survey introduces a selection bias (older people, low income people, probably North and NorthEast Brasil) with lower social media adherence. Likewise, it needs to be made clearer that this is NOT a representative survey.
Answer: Thanks for pointing this out. Please note that we have revised the discussion section of the new version of the paper, and added other limitations of the study, including that we did not use a probability sample.
- What was the attrition between people who were first interested but did not meet the inclusion/ exclusion?
Answer: This is a very important point, but unfortunately, we were not able to compare demographic characteristics between those who completed the questionnaire and those who did not. We believe that many people accessed the questionnaire just out of curiosity, did not complete it, and the demographic characteristics reported are therefore not reliable.
- You describe the Q with respect to children - what was the Q for adults only?
Answer: Please note that in the new version of the manuscript, we have rewritten the objectives in a way that may be more understandable to readers.
- It is unclear and needs to be justified why the lower age for inclusion in the survey was 18 years and why there is a gap in assessement between 12-18 years (you only talk about kids below 12 yoa and then you have a lower age of 18 y). Adolescents are an important multiplier group and apt to social media. This is a missed opportunity.
Answer: We agree with the reviewer's assessment. However, in Brazil, according to the current resolution of the National Health Council (Resolution 466/2012), research involving children under 18 must have a Free and Informed Consent Form signed by the parents or legal representative and the Free and Informed Assent Term signed by the person under 18 years of age. This would not be a simple task, although the information obtained would be of great importance.
- you use the term "children hesitancy". In fact, as you did not ask children, you need to modify this term to something like "parent's hesitancy towards children vaccination"
Answer: This is an excellent commentary. We clarified this issue in the manuscript.
- It would have been beneficial if the authors had differentiated between hesitancy (which is well explained) and rejection: how many of the "No's)" are indeed hesitaters and how many are plain rejectors.
Answer: This is another important point. When we planned our study, we thought about including rejection in the instrument, but it seemed difficult to formulate a question avoiding any polemics. Therefore, we also added a comment about this among the limitations of the study in the discussion section of the revised version of the paper.
- Tbl 2: it is problematic to single out the Federal government: Brasil as a federal union allows also the states quite some autonomy to set up policies incl vaccination priorities and choice of vaccine (Sao Paulo!).
Answer: The reviewer is correct, and we have also added this comment among the limitations of the study.
- There was a survey done by the government about paeds vaccination, The outcome should be mentioned and referenced.
Answer: We have added comments on this research in the third paragraph of the introduction.
- Whilst the mortality data in Brasil are correctly described and discussed, it is incomplete as the excess mortality data which is similar to ie US or Italy should also be described and referenced.
Answer: An article published in The Lancet showed that, at the country level, the highest numbers of cumulative excess deaths due to COVID-19 between 2020 and 2021 were estimated in India (4.07 million [3.71–4.36]), the USA (1.13 million [1.08–1.18]), Russia (1.07 million [1.06–1.08]), Mexico (798 000 [741 000–867 000]), Brazil (792 000 [730 000–847 000]), Indonesia (736 000 [594 000–955 000]), and Pakistan (664 000 [498 000–847 000]). Among these countries, the excess mortality rate was highest in Russia (374.6 deaths [369.7–378.4] per 100 000) and Mexico (325.1 [301.6–353.3] per 100 000), and was similar in Brazil (186.9 [172.2–199.8] per 100 000) and the USA (179.3 [170.7–187.5] per 100 000) (https://doi.org/10.1016/S0140-6736(21)02796-3). Although these numbers are important in highlighting Brazil as one of the countries with the highest numbers of cumulative excess deaths due to COVID-19, we could not decide on the best way to cite this information in our article briefly without taking the focus of the vaccination. We are open to any suggestions.
- line 278: sentence incomplete
Answer: We corrected this mistake in the revised manuscript.
- line 316: there are no data in your survey which justify this statement. Pls avoid politicising your work. This is all not black and white - whilst part of the government indeed voiced rethorics against vaccination, at the same time Brasil acquired over 680 Mio doses. And the coverage is among the highest in large size countries. What also did not help was the attitude of some governors and one manufacturer who made public flawed data and criticised organisations such as Anvisa which is perceived internationally as one of the agencies who best mastered the pandemic.
Answer: The Brazilian government threatened Anvisa officials, strongly questioned the vaccine's safety, created a public consultation with poorly formulated and biased questions, and only on 5th January 2022 the final government's approval was granted (doi 10.1016/j.lana.2022.100246). Contrary to the decision, the presidential rhetoric continued to instill fear, widely publicising that he would not vaccinate his 11-year-old daughter (doi 10.1016/j.lana.2022.100246). Brazil could have been the first in the world to start vaccination if the federal government had cooperated. The president's declarations against the vaccine left negotiations on hold and delayed the start of vaccination in the country. If Brazil bought millions of doses of the vaccine and obtained efficient vaccination coverage of the population, this happened because of a Unified Health System that was already established in the country, with professionals prepared for the crisis, and with the support of the academic, private, and some political sectors. While all this was going on, the president of the republic offered chloroquine to an emu, aiming to ridicule all the sectors that were looking for a solution to the crisis.
The first offer of the Coronavac vaccine to the Ministry of Health was on July 30, 2020. A new offer was made on October 7, 2020. On October 19, finally the Ministry of Health expressed its intention to buy the vaccines. However, two days later, the president of the republic said he would not buy the vaccines. On January 17, the vaccination campaign is started in São Paulo, after emergency authorization from Anvisa. Meanwhile, many people were dying every day from COVID-19. The attitude of some governors and of a manufacturer that made flawed data public and criticized ANVISA does not seem to us as relevant as these omissions by the federal government, considering that the president of the republic has greater potential as an opinion maker with the population.
Therefore, it is impossible today to write a scientific paper about the perceptions, attitudes, and practices of the Brazilian population about COVID-19 without referring to the attitudes of the federal government during the pandemic. These discussions are not associated with a left-wing or right-wing political position, but rather with a decision of whether we will defend science or pretend that the scientific denialism promoted by the federal government is not affecting the overall health of the population. In conclusion, we do not believe we are politicizing our work. All the comments in our discussion section are based on the bibliographic references cited, which do not exhaust the examples of the federal government's negligence in combating the pandemic.